# Quantity and Distribution of Muscle Spindles in Animal and Human Muscles

**DOI:** 10.3390/ijms25137320

**Published:** 2024-07-03

**Authors:** Yunfeng Sun, Caterina Fede, Xiaoxiao Zhao, Alessandra Del Felice, Carmelo Pirri, Carla Stecco

**Affiliations:** 1Padova Neuroscience Center, University of Padova, 35129 Padova, Italy; yunfeng.sun@studenti.unipd.it (Y.S.); xiaoxiao.zhao@studenti.unipd.it (X.Z.); alessandra.delfelice@unipd.it (A.D.F.); 2Institute of Human Anatomy, Department of Neuroscience, University of Padova, 35122 Padova, Italy; caterina.fede@unipd.it (C.F.); carmelo.pirri@unipd.it (C.P.); 3Section of Neurology, Department of Neuroscience, University of Padova, 35122 Padova, Italy

**Keywords:** muscle spindle, fascia, distribution, connective tissue, proprioception, density

## Abstract

Muscle spindles have unique anatomical characteristics that can be directly affected by the surrounding tissues under physiological and pathological conditions. Understanding their spatial distribution and density in different muscles is imperative to unravel the complexity of motor function. In the present study, the distribution and number/density of muscle spindles in human and animal muscles were reviewed. We identified 56 articles focusing on muscle spindle distribution; 13 articles focused on human muscles and 43 focused on animal muscles. The results demonstrate that spindles are located at the nerve entry points and along distributed vessels and they relate to the intramuscular connective tissue. Muscles’ deep layers and middle segments are the main topographic distribution areas. Eleven articles on humans and thirty-three articles on animals (totaling forty-four articles) focusing on muscle spindle quantity and density were identified. Hand and head muscles, such as the pronator teres/medial pterygoid muscle/masseter/flexor digitorum, were most commonly studied in the human studies. For animals, whole-body musculature was studied. The present study summarized the spindle quantity in 77 human and 189 animal muscles. We identified well-studied muscles and any as-yet unfound data. The current data fail to clarify the relationship between quantity/density and muscle characteristics. The intricate distribution of the muscle spindles and their density and quantity throughout the body present some unique patterns or correlations, according to the current data. However, it remains unclear whether muscles with fine motor control have more muscle spindles since the study standards are inconsistent and data on numerous muscles are missing. This study provides a comprehensive and exhaustive approach for clinicians and researchers to determine muscle spindle status.

## 1. Introduction

The intricate physiology of skeletal muscles involves myriad sensory structures, among which muscle spindles (MSs) emerge as a pivotal player [1]. As highly specialized proprioceptive organs, MSs serve as sensory receptors crucial for motor control and muscle tone maintenance. Knowledge of MS distribution also impacts neuromuscular rehabilitation [2,3]; studies [4,5] proved that the center of the highest region of muscle spindle (MS) abundance (CHRMSA) should be the primary injection point of botulinum toxin in muscle hypertonia. Increasing evidence has demonstrated that MSs have a strong reactive relationship with surrounding tissues. The modification of surrounding tissues will affect the function and sensitivity of MSs and result in the eventual aggravation of movement performance [6,7,8]. Understanding different muscles’ spatial distribution and density is paramount for unravelling the complexities of motor function and holds profound implications for clinical and biomechanical studies.

This review focuses on MS distribution within skeletal muscles as well as the density and quantity of MSs in different muscles. We also report the bottlenecks in MS research to contribute to the advancement of knowledge in the field.

### Muscle Spindle Neurophysiology

MSs are the principal kinesthetic receptors in mammalian skeletal muscles: they inform the central nervous system about muscle fiber length modifications and contribute to motor control and the body’s sense of positioning [9]. MSs are located in almost all muscles and are attached to extrafusal muscle fibers in a consistent direction [10]. It was suggested that there are approximately 50,000 MSs in the entire human musculature [11]. MSs are spindle-shaped. In the central region of the muscle spindle, sensory nerve terminals are observed to connect with intrafusal fibers, and the diameter of spindle capsule is increased in this area [12,13]. A complete MS includes intrafusal fibers, vessels, and innervation. All these structures are surrounded by capsules and linked together to muscle fiber via connective tissue [13,14]. The human intrafusal fibers can be up to 8 mm long and are pooled in groups of 8–20. In mice, their length reaches up to 400 μm, and they are clustered in groups of three to five [15,16]. There are three types of intrafusal fibers in mammalian spindles: large nuclear bag 1 fibers, larger nuclear bag 2 fibers, and smaller nuclear chain fibers. These fibers are thinner than extrafusal muscle fibers [17]. Nuclear bag fibers extend beyond the capsule and are attached to intramuscular connective tissue (IMCT); chain fibers link with the polar of nuclear bag fibers [16,17].

Mammalian MSs have afferent and efferent nerve integrated innervation; the type Ia and type II nerve fibers are the dominant sensory innervation (afferent) and have different axonal conduction velocities in humans [18,19]. Type Ia afferent nerve terminals form annulospiral endings by reshaping irregular coils and linking with the area around the equatorial region (the middle part of MS is also the thickest). Type II nerve terminals are embedded on the one side of type Ia fibers’ annulospiral ending [15,18]. γ-motoneurons are the efferents of intrafusal fibers; some intrafusal fibers are also innervated by β-motoneurons, which connect to the extrafusal [13,20]. Axons of efferent and afferent fibers enter the spindle through the outer membrane at the equator region; the efferent motoneurons separate and connect to the polar regions [16,20]. γ-motoneurons can be divided into dynamic and static fibers [20]; dynamic γ fibers are connected to nuclear bag 1 fibers and static γ fibers are connected to nuclear bag 2 fibers and chain fibers [18,21].

In general, MS is a muscle fiber stretch sensor that measures the degree and speed of muscle length changes. The spindles’ intrafusal fibers extend when related muscle fibers are stretched since the spindle intrafusal and muscle fibers run in the same direction [22]. It is well established that action potentials will be generated in afferent sensory neurons according to the degree and rate of stretching [23]. Type Ia afferent nerve terminals form annulospiral endings (primary ending), innervate all intrafusal fibers, and exhibit peak responsiveness to variations in muscle fiber length and muscle-stretch magnitude [21]. Afferent sensory neurons of type II (second ending) innervate the nuclear bag 2 fibers and chain fibers, and exhibit maximal activity in response to muscle-stretch magnitude [18]. Afferent sensory action potentials are generated in the target and antagonistic muscle due to the constant coordination of movements [24]. For a summary, see Figure 1.

## 2. Methods

We followed the guidelines for narrative reviews [25].

### 2.1. Searching Strategy for MS Distributions

#### 2.1.1. Databases and Terms

The databases used included Web of Science (WOS), PubMed, and Cochrane, with the keywords of muscle spindle*, distribut*(distribution; distribute), and locat*(location; locate), following the string PubMed: (muscle spindle*[Title/Abstract]) AND (distribut*[Title/Abstract] OR locat*[Title/Abstract]); WOS: (muscle spindle*[title]) and (distribut* or locat* [topic]); Cochrane: (distribution or location or distribut* or locat*[Title Abstract Keyword]) and (muscle spindle*[Title Abstract Keyword]).

#### 2.1.2. Inclusion and Exclusion Criteria

All relevant articles published by 22 March 2024 were included. The exclusion criteria were articles not written in English, non-original research articles (review articles, letters or comments, books, abstract only), and duplicates. A total of 734 papers were retrieved (PubMed: 573, WOS: 150, Cochrane: 11). A total of 701 articles were written in English, and 144 articles were excluded as duplicates. A total of 557 articles were retained.

Among these 557 articles, 13 full texts were inaccessible due to invalid links in the database or withdrawal. Another 12 non-original articles were excluded. After title, abstract and full-text screening, 56 articles were eventually included in the final analysis (details in the flowchart in Figure 2).

Two authors (Y.S. and X.Z.) identified and executed the search strategy. A third author (C.F.) resolved any disagreement that arose regarding inclusion/exclusion and data extraction.

A predefined data extraction sheet including “muscle name, MS study method, species and distribution area (deep to superficial layer; proximal to distal region; related to which special anatomical structures)” was collated.

### 2.2. MS Quantity Search Strategy

#### 2.2.1. Databases and Terms

The databases included WOS, PubMed, and Cochrane. The search keywords included muscle spindle*, number*, amount, and density, and they follow the terms (muscle spindle*[title/abstract]) and (number* or amount* or density [title/abstract]) in PubMed; “(muscle spindle*[title]) and (number* or amount* or density [topic])” in WOS; and “title/abstract/keyword” in Cochrane.

#### 2.2.2. Inclusion and Exclusion Criteria

All relevant articles published by 22 March 2024 were included. The exclusion criteria were as follows: non-English writing, without full-text access, non-original articles (review article, letters or comments, books, abstract only), and duplicates. A total of 716 articles were found in Cochrane/PubMed/WOS (12, 572, and 132, respectively). Among those 716 articles, 677 were written in English. Among those 677 articles, 545 were selected after being checked for duplicates. Two books, four review articles, and thirteen non-full-text articles were excluded. Then, 526 articles remained. After title, abstract, and full-text screening, 26 articles were included. Upon reviewing the articles included in “Section 2.1”, we identified 18 articles that met the requirements. In the end, forty-four articles were included (Figure 3).

Two authors (Y.S. and X.Z.) identified and performed the search strategy. They selected the included studies. A third author (C.F.) resolved any disagreement that arose regarding inclusion/exclusion and data extraction.

A predefined data extraction sheet including “species, muscle name, muscle mass, number of MSs, density, references, stain” was collated.

## 3. Results

### 3.1. Studies in Animals and Humans

Regarding distribution, 56 studies were included. A total of 90 muscles (62 types) were investigated in 20 species, including humans, rodents, cats, and avians. Most articles focused on humans (13 articles), rodents (12 articles) and cats (10 articles). The masseter and temporal muscles were the most frequently evaluated. Studies on distribution in those 90 muscles mainly examined humans (21 samples), rodents (17 samples), and cats (13 samples). Details are presented in Appendix A.

Regarding number and distribution, 44 articles were included: 11 articles on humans, 8 on cats, and 6 on rodents (see Appendix A). A total of 189 animal muscles and 77 human muscles were investigated; the details are presented in Appendix A. A total of 24 animal species were involved, with rodents and cats being the most popular subjects (33 and 39 reports, respectively) (see Appendix A). The soleus and masseter were the main targets among the total 189 animal muscles (see Appendix A). The full data on those 189 muscles could not be extracted due to incomplete reports. The quantity of MSs was deduced successfully in 186 muscles, while 134 had muscle weight data; 131 muscle samples were simultaneously evaluated both in terms of muscle weight and MS quantity. In humans (see Appendix A), a total of 77 muscles (36 kinds) were investigated.

### 3.2. Muscle Types Studied

Diverse human and animal muscle types were analyzed according to the article results. In humans, the most frequently studied muscles were the pronator teres/medial pterygoid muscle/masseter/flexor digitorum and other hand or head muscles. Regarding animals, the most frequently studied muscles were the soleus (10 times), masseter (12 times), and temporal (9 times). Other muscles that were frequently studied included the extensor digitorum communis/medial pterygoid/extensor pollicis, with frequencies of 6/7/6, respectively. As shown in Appendix A, the studied animal muscles included all anatomical districts, but these were few in humans.

### 3.3. Analysis of Staining Methods

We summarized the staining methods. In the 82 articles (all articles included in the present study), silver staining and hematoxylin and eosin (H&E) staining were the main MS research methodologies. Silver staining was used more than 70 times in animals and 2 times in humans. H&E staining was used 57 times in 60 studies on animal and human muscles. Moreover, other methods were applied, as follows: modified Sihler’s staining and the X-ray reading box were used to identify the intramuscular nerve-dense region, and then the number of MSs was counted after the sample was cut into sections, and H&E staining was performed. Another novel method is synchrotron radiation-based computed tomography (SRCT), which can generate a 3D visualization of and MS quantification for related nerves and vessels. The details are presented in Figure 4.

### 3.4. Density and Quantity in Different Muscles

Within currently collected animal muscles, the superior oblique of pig was found to have the most MSs with 310.30. No spindles were found in the adductor mandibulae externus profundus and musculus adductor mandibulae posterior in Anas platyrhynchos (Figure 5 and Appendix A).

In humans, a total of 77 muscles (36 kinds) were investigated. Data on muscle mass were obtained from 59 of those 77 muscles, and the MS quantity was completely extracted. We present the muscle data according to different parts of the human body. In humans, muscles from the head, upper limbs, and neck were mainly studied, with an underrepresentation of other areas (Figure 5).

In animals, the results of relevant animal studies are presented separately by species. Muscles in each species are presented in order of MS quantity (Figure 5). Pig—superior oblique/cat—complexus/pig—superior rectus/pig—medial rectus/pig—inferior rectus/pig—lateral rectus had the largest number of MSs (310.30/254.00/211.60/208.40/191.70/190.00). Regarding humans, trapezius/latissimus dorsi/platysmata/lateral pterygoid/longus colli/multifidus were the muscles with the most MSs, numbering 437.00/368.00/177.00/155.00/143.00/111.00. As shown in Figure 5, trunk or head muscles contain many MSs compared to muscles in the area of the body. During the literature search stage, we found another related review article containing information on human muscle MSs [26]: Longissimus capitis, 507.00 (number of MSs); Semispinalis capitis, 619.00; Longissimus dorsi, 1193.00; Obliquus externus abdominis, 569.00; Pectoralis major, 450.00; Iliocostalis, 796.00; Triceps brachii, 520.00; Iliopsoas, 656.00; Adductor magnus, 437.00; Vastus lateralis, 449.00; Gluteus maximus, 629.00. Data presented in the article (the related review article, reference No.26) failed to reference the sources. In addition to references, most human data were quoted from another study (study samples were obtained from a human newborn cadaver) which was written in German. However, we failed to access the full text of the German article. Therefore, the data involved in the article [26] were not presented and analyzed in the present study. We must clarify this issue to ensure this study’s objectivity. The collected human data (Appendix A) showed a statistical correlation between muscle mass and the MS quantity. However, we suggested that this data calculation method was unreasonable and non-scientific as some standards were disregarded and data were incomplete, as mentioned in the “Section 4”.

Regarding density, the most common methodology was the MS quantity per gram of muscle (n°/gram). Data showed that the M. coracotriceps of pigeons (the results of a five-sample study) had the highest overall density, with a mean value of 14582.00 per gram, ranging from 12222.00 to 16667.67 (Figure 5). In humans, the inferior oblique/superior oblique/rectus capitis posterior has the highest density compared to other investigated muscles, with values of 266.67/189.47/98.31, respectively (Figure 5). Another interesting finding was that the muscles in the arm and head–neck area apparently have the highest MS density based on the observation of existing collected data; this proposal seems similar to the original hypothesis that “muscles with high densities are primarily involved in fine motor control”. However, we have reservations, which are detailed in the “Section 4”; a study on cat complexus muscles showed two different results amounting to MSs 254.00/190.00 while the density was 106.72/71.43. This was one reason why we regard the above conclusions with reservations, and there were many reasons (method or counting) for the data discrepancy, as mentioned in the “Section 4”. Another study [27] about muscle architecture and MS abundance also expressed reservations regarding the above hypothesis. Moreover, MS density was referred to as both “number/gram”, while another term, “number/section area”, was used in other studies; this method had the same steps, including sample preparation, slice, and staining followed by MS counting and section area calculation. The term “/gram” refers to the muscles’ MS integrity, while the “/section area” represents density in different muscle parts (regions).

### 3.5. Distribution and Location Related to the Muscle Fiber, Vessel, NEPs (Nerve Entry Points), Fascia, and Other Anatomical Structures

We identified some typical MS distribution patterns. Special structures, including NEPs, vessels, fascia, and special muscle regions, are the main distribution areas. A huge number of studies proved that MSs are related to nerve entry points.

First, the MS distribution [28] was found to be diverse in different muscle fiber types. Type I and type II muscle fibers co-exist to exert diverse biological functions: the proportion can be modified by disorders and different influencers [29]. Conclusions from the animal studies, including the cat flexor carpi radialis muscle, turtle ambiens muscle, guinea pig medial gastrocnemius/vastus lateralis, and the rat sternomastoid muscle, suggest that most MSs are located in slow muscle fiber (type I). However, this remained unclear in humans. The unique location of the MSs may suggest that they can help postural control, but the mechanism of this phenomenon was not explored, and muscle fiber type was not a common marker of MS localization in MS research.

Beyond muscle fibers, the nerve branch direction is also related to MS distribution [13]. The research results derived from 26 muscle samples across 12 studies strongly suggested that MSs were distributed following the NEPs and nerve branch direction. The NEPs were mainly described as those histological sections on which the main nerve trunk(s) burrowed deep into the epimysium [30]. In mice, research on the extensor digitorum longus and soleus muscles demonstrated that the MS distribution map regularly followed the location area of the NEPs [31]. The results were consistent with Valdez’s studies [32] as well dromedarius (intraorbital skeletal muscles), cat (rectus femoris, tibialis anticus muscles), and avian (anterior and posterior latissimus dorsi muscles) study conclusions. In humans, research conclusions about the intrinsic postvertebral muscles, suboccipital muscle, and forearm muscle also suggest that MSs are located predominantly at the NEPs (Appendix A). On the contrary, research based on mice’s lower extremities has suggested that MSs are close to the NEPs in the extensor digitorum longus and soleus muscles but not the lateral gastrocnemius [32]. Although some differences existed, most studies indicated that NEPs are one of the main distribution regions. It is important to note the structural differences in neuromuscular junction between animals and humans. A previous study showed that the neuromuscular junction in humans was much smaller and more limited [33]. This difference may suggest that MSs in human muscles are much more concentrated than in animal muscles, but no study has clarified the relationship between the neuromuscular junction and MSs. The present study also failed to clarify this issue.

Evidence on MS distribution related to the blood vessels is currently not conclusive. A study [34] based on mice extensor digitorum longus followed by ATPase staining demonstrated that MSs were close to vessels. Another investigation [35] on human suboccipital muscles using silver staining also showed that most spindles were detected among major blood vessels and nerve fibers. A similar conclusion was also reached in hamster tenuissimus muscle. The authors used H&E and ATPase staining in three suboccipital muscles, proving that spindles embedded within a matrix of perimysium connective tissue and the main muscle artery and vein were located centrally within a closely packed array of extrafusal fibers [36]. The vessel may be an MS localization marker, but this lacks strong evidence.

In addition to NEPs and vessels, MSs are also present in other structures. Research has proved that MSs are connected to intramuscular connective tissue (IMCT) and affected by IMCT activities [37,38]. MSs are embedded in the perimysium in rat sternomastoid muscle [39]. It is also well established that 59.2% of MSs in sheep multifidus muscle are close to a major fascial element [40]. Another research work [8] also suggested that the MS’s outer capsule is continuous with intramuscular connective tissue. Some earlier studies in Japanese shrew-mole with H&E staining also showed that MSs were strongly connected to IMCT in medial pterygoid and masseter muscles [41]. The above studies provided a novel insight, showing that considerable amounts of MSs are close and connected to IMCT. However, further investigation is needed to clarify the interactive relationship between IMCT (fascia) and MS in humans. For details, see Appendix A.

### 3.6. MSs Topographic Distribution in Muscle

In addition to special anatomical structures, the distribution and main MS concentration area in muscle also show some unique characteristics. The results from studies on 32 muscles demonstrated that the middle region had the most MSs compared to the proximal and distal regions. A study on 25 muscles revealed that the deep layer contained the most MSs compared to the superficial layers. The proximal, distal, and superficial distribution regions had sample sizes of 27, 14, and 14 muscles, respectively, in animals and humans. Even though this was the conclusion made by most researchers, differences remained. For example, a study [42] on cat masseter muscles revealed that the proximal and distal regions had the highest amounts of MSs. Another investigation on cat masseter suggested that MSs are located in the distal region [43]. Interspecies comparison between the platysma muscles in humans and rhesus monkeys showed that MSs were mainly located in the proximal region in monkeys and the proximal and middle areas in humans without statistical significance [44,45]. Based on the collected data results, MSs are mainly located in the deep layer and middle region of the muscles, followed by the superficial layer and proximal regions, while the distal regions contain the fewest MSs (Appendix A and Figure 6).

## 4. Discussion

Muscle spindles are the main proprioception organ, and they play an essential role in motor control and musculoskeletal functions. In the present study, the distribution and average MS quantity in animal and human muscles was reported. Since our research intention was to map MS distribution and quantity in specimens from healthy subjects, these results may not be applied to pathology, but the MS changes in the pathological state mainly include physiological function and morphological characters, not the distribution and amounts [45,46]. Hence, our study also had guiding significance for the MS research under disease conditions.

As for distribution, the included MS distribution studies often failed to provide quantity and density data, while MS distribution patterns were not investigated in most MS quantity and density studies. We stress that MS distribution is related to specific anatomical structures and muscle regions. Differences remain in different muscles; the distribution pattern is typically NEPs/vessel/IMCT. A common pattern regarding muscle portions is deep > superficial/middle > proximal > distal.

Regarding quantity and density, the MS quantity appears to be highest in trunk muscles or other big muscles in humans, while the muscles in the neck and head, such as oblique muscles and rectus capitis posterior, have the highest density. Data on lower limb muscles and other big muscles are still missing. In animals, the most MSs were found in pig oblique and rectus muscles, and no MS was found in the adductor mandibulae externus profundus of anas platyrhynchos. The highest density was found in pigeon M. coracotriceps, with no correlation between MS quantity and muscle mass in animals. The studied animal muscles were found to be widely distributed in all areas of the body. A correlation test between human muscle mass and MS quantity showed a positive correlation, but we have reservations about this observation: First, the lack of unified standards for muscle tissue acquisition and the quality assessment process have led to differences in muscle quality measurement results. This may cause data inconsistencies. Second, there are methodological drawbacks, i.e., the test is unable to provide the exact MS quantity in muscle by cutting muscle into sections (thickness, numbers, count method, and others). Most importantly, many types of human muscles have not been studied. Given these reasons, current data in the literature were unable to clarify this issue. It is suggested that a uniform experimental process will promote the integrity and reliability of MS studies.

After comprehensively reviewing the data collected about the MS density, quantity, and distribution, some ambiguity remains. First, most studies about muscle density and quantity failed to identify which part of muscle has the highest density/number or which related structures have the most MSs, focusing instead on the ratio of MS quantity and muscle weight. It is well established that understanding which part or area has the highest MS density/quantity will contribute to guiding clinical practice, such as promoting the efficiency of botulinum toxin injection or the MS-based treatment of musculoskeletal disorders. Second, based on the current data, we also failed to clarify whether muscles with fine motor control have the highest number/density of MS. We identified the following areas of research that need to be addressed: To provide a precise count of MSs, a consensus on methodology needs to be reached. Topics to be addressed include how to slice different muscles, how many slices should be made in different muscles, and the standards and procedures of collecting and disposing of muscle samples. Data on muscles in other body areas present an urgent need in humans. Another underexplored topic is how MSs interact with other proprioception/pressure sensors, such as the Golgi tendon organ [47] and Ruffini’s corpuscles [48]. In consideration of this physiological integration, we need to consider that proprioception is not solely controlled by MS, a fact also confirmed by the lack of MS in the adductor mandibulae externus profundus of anas platyrhynchos [49]. It is challenging to describe the exact relationship between MS and proprioception without considering other receptors.

Considering the methodology used in MS studies, staining with H&E, silver, and others are the standard methods. These techniques are simple, easy to perform, and widely applicable. Some novel techniques (SRCT) have advantages, such as better time and labor efficiency and higher accuracy and quality, but these are still not widely adopted. 

Limitations remain; even the present study followed the relevant principles in the literature search and data extraction and analysis: the inclusion and exclusion criteria were relatively strict, which might lead to some articles not being included. In particular, the language limitation led to the absence of some non-English articles, which might have an impact on the conclusions of the present study. Second, the present study failed to classify and analyze the effects of different MS research methods (staining methods, counting methods, etc.) on the results regarding the number and distribution of MSs. Considering the amount of muscle data available at present, it is not possible to provide a scientific and comprehensive answer to this question.

## 5. Conclusions

According to the results of current studies, the deep layer and middle region are the main distribution areas of MSs, followed by the superficial, proximal, and distal areas of the muscle. In addition to muscle region division, several special anatomical structures, including NEPs, vessels, and IMCT, are the structures most commonly associated with MSs. The number and density of MSs are still not clearly defined due to the lack of data on lower limb muscles and other big muscles in humans.

Based on our results, we cannot confirm that muscles that serve postural control and fine movement have a higher density of MS. Lastly, we propose a set of operating standards for MS research.

## Figures and Tables

**Figure 1 ijms-25-07320-f001:**
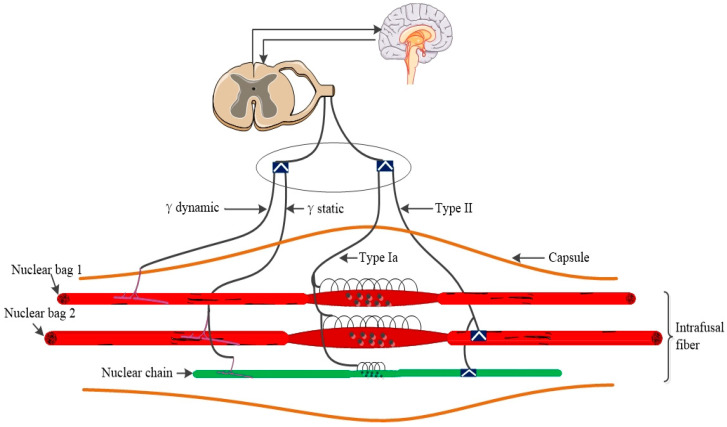
Basic structure of MS; nuclear bag fiber is thicker than chain fiber, with nuclear bag 2 fiber being the thickest overall. Afferent sensory neurons of type Ia located in the area of equatorial region contribute to innervating all fibers. Afferent sensory neurons of type II fibers located to the side of Ia, innervate the nuclear bag 2 fibers and chain fibers. Signals generated from mechanical force sense turn into electro-neural signals and ascend into the brain via the spinal cord, contributing to proprioception and motor control.

**Figure 2 ijms-25-07320-f002:**
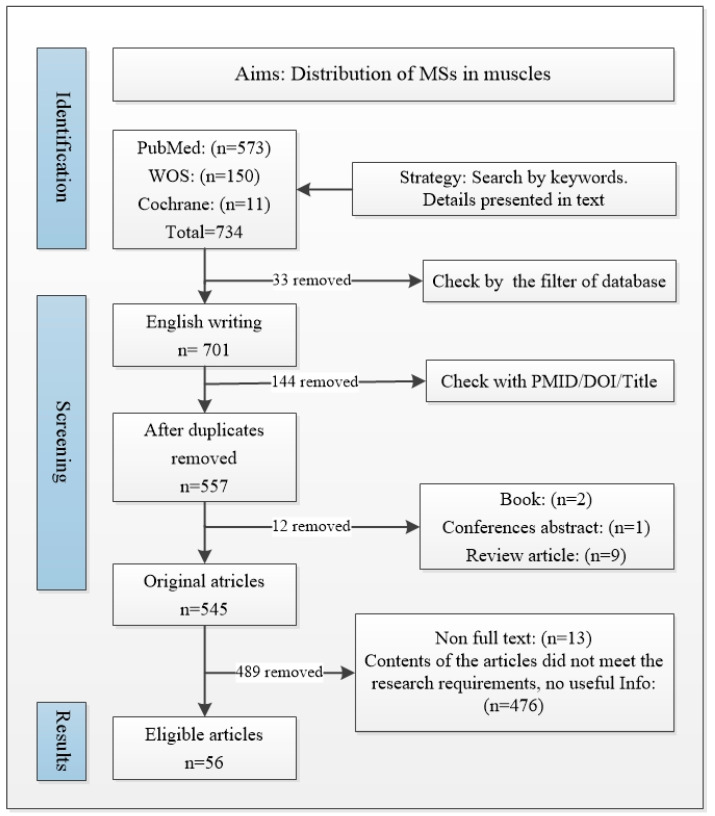
Flow diagram of search strategy for MSs distribution.

**Figure 3 ijms-25-07320-f003:**
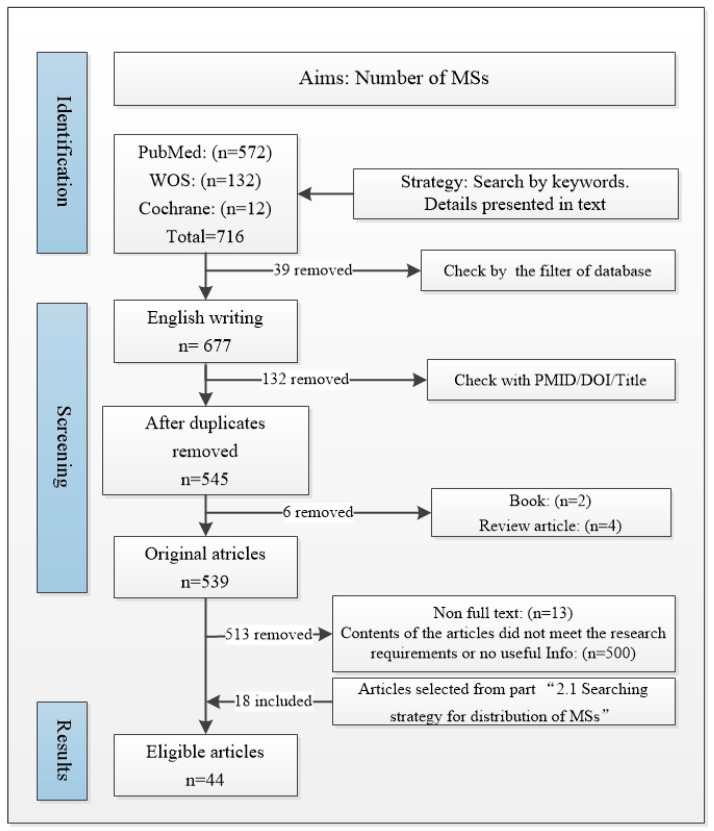
Flow diagram of MSs quantity/density.

**Figure 4 ijms-25-07320-f004:**
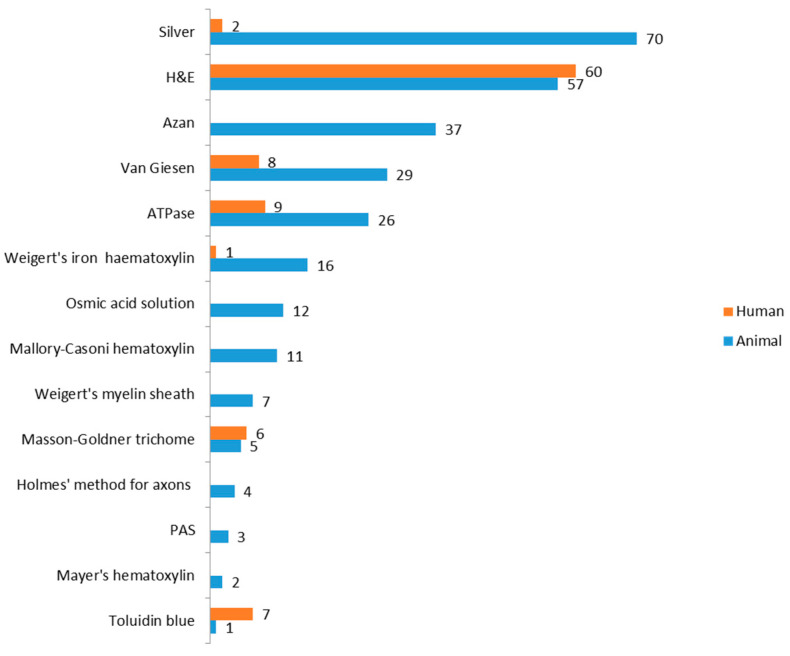
Staining methods used in MS studies. Silver and H&E were the main techniques. The bars in blue present the animal studies and orange bars stand for human studies. PAS: Periodic Acid-Schiff stain.

**Figure 5 ijms-25-07320-f005:**
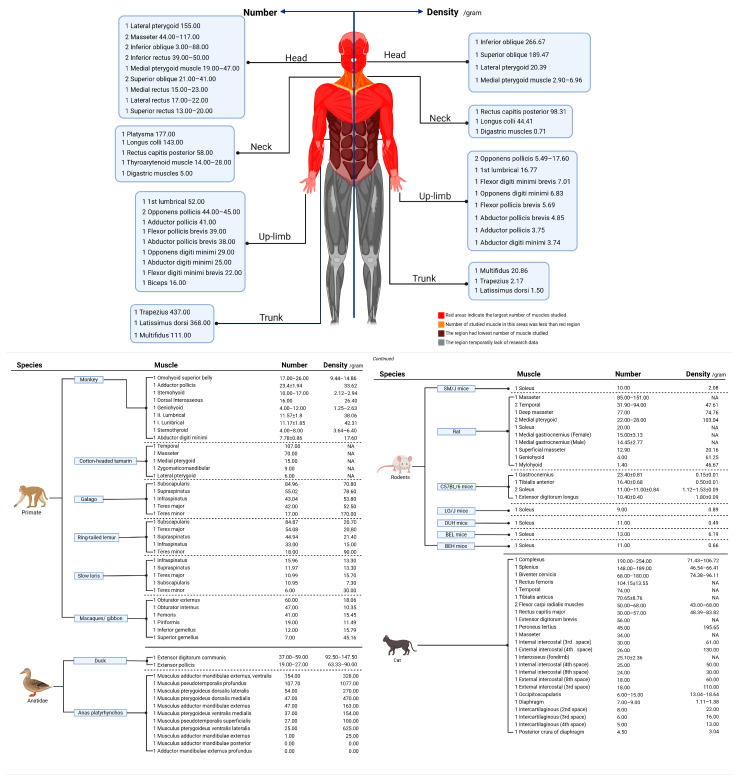
Number and density of MSs in human muscles and animal muscles; density of MSs presented in “n°/ gram”. Some data of animal muscles and human muscles were not presented due to space issue. For more details, please check the Appendix A. The number in front of muscle name indicates the number of studies reporting the finding.

**Figure 6 ijms-25-07320-f006:**
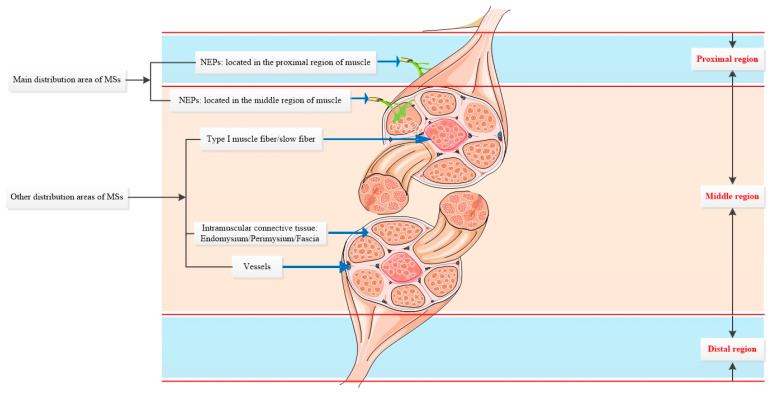
The main distribution area of MSs, a piece of muscle cut laterally was used for presentation. NEPs were the main related anatomical structure followed by vessels, IMCT and type I muscle fiber. From the perspective of the anatomical position of the proximal and distal region of muscle, the middle region owned the largest number of MSs in animals while proximal region owned the largest number in humans.

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
