# Peer review of "Quantity and Distribution of Muscle Spindles in Animal and Human Muscles"

_ijms, 2024, doi:10.3390/ijms25137320_

Round 1
Reviewer 1 Report
Comments and Suggestions for Authors
The authors aim to construct a comprehensive catalogue of the literature on the numbers of muscle spindles in skeletal muscles, and their distribution within muscles in relation to other components of muscles, including blood vessels and intramuscular nerves. Where relevant they also attempt to demonstrate the clinical relevance of this information.
The Introduction is intended as a brief overview of muscle spindle structure and function, but much of it is unclear, either due to difficulties with English, or to misunderstanding of the literature.
One of the main categories of data collated is the so-called “spindle density” or number of muscle spindles per gram of muscle mass, for anatomically identifiable muscles. It is then considered uncritically as an unbiassed measure of relative abundance of muscle spindles, not only in different muscles from a single species, but also in muscles of different species including non-mammalian species. The deficiencies of spindle density used in this way were demonstrated by Banks in 2006 (J. Anat. 208: 753-768). Even more surprising, perhaps, is the omission of the data on some 230 human muscles collated by Voss in 1971 (Anat. Anz. 129: 562-572).
Specific points:
L 13 “never” to read “nerve”
L 14 abbreviation IMCT not defined
Lines 44-48 unclear
L 64 what does “greater biological characteristics” mean?
Lines 67-69 no identifiable source is given for the conduction velocities, which are certainly wrong.
L 74 what does “the same arrow direction and anatomical characteristics” mean? Intrafusal and extrafusal fibres are certainly not anatomically similar.
Lines 76-79 Primary endings, as noted elsewhere in the paper, usually innervate all three types of intrafusal fibre, and consequently show both dynamic and static sensitivity.
L 82 Phospholipase D isn’t an ion channel.
L 84 the spinal cord is an integral part of the central nervous system.
L 98 does PRISMA require explanation and a reference?
Lines 108 and 136 delete “respectively” to make sense.
Lines 151-2 Does “A total of 62 types of 90 muscles” mean “A total of 90 exemplars of 62 different muscles”?
L 170 and elsewhere. Does “pieces of muscle” mean “individual muscles”?
In the legend to fig 4 do “results of part 2.1” and “results of part 2.22 refer back to the Methods? If so, there is no 2.2, only 2.1.2. Also F and G are confused.
Lines 243-245 ‘This proposal seems similar to the original conclusion of “subjective descriptions of fine motor control and brought out the concept that the density of MSs is higher in those muscles which own precious control of movement” [25].’ This appears to be a quotation from ref 25, but the sentence does not occur in that reference.
References: while this paper was with the reviewer, another relevant paper on the distribution of spindles in human deep cervical muscles has been published by Wang et al. Frontiers in Neuroanatomy https://doi.org/10.3389/fnana.2024.1340468
Comments on the Quality of English Language
As indicated above, extensive editing of the English is needed. There are many examples where it is difficult or impossible to even guess at what the authors intend.
Author Response
Dear reviewer:
First and foremost, we would like to express our deepest gratitude for your thorough review of our manuscript. Your insightful comments and suggestions are invaluable to us, and we appreciate the time and effort you have dedicated to providing this feedback. We read all carefully and revised our manuscript based on the precious advice. Thank you so much.
The following content in italics (labeled with ★) is your valuable suggestions, text in red is our correction.
★The Introduction is intended as a brief overview of muscle spindle structure and function, but much of it is unclear, either due to difficulties with English, or to misunderstanding of the literature.
Answer: Dear reviewer, thank you in advance. We are sorry for the mistakes in language and those related misunderstanding. We have revised the main text of “Introduction”, also the English writing issue has been edited by MDPI.
★One of the main categories of data collated is the so-called “spindle density” or number of muscle spindles per gram of muscle mass, for anatomically identifiable muscles. It is then considered uncritically as an unbiassed measure of relative abundance of muscle spindles, not only in different muscles from a single species, but also in muscles of different species including non-mammalian species. The deficiencies of spindle density used in this way were demonstrated by Banks in 2006 (J. Anat. 208: 753-768). Even more surprising, perhaps, is the omission of the data on some 230 human muscles collated by Voss in 1971 (Anat. Anz. 129: 562-572).
Answer: Dear reviewer. We are grateful for this precious advice, thank you for giving us such valuable advice. We read the mentioned paper carefully, term of “density” we used in this manuscript was not the best way to measure the abundance of MS in muscle. By using “residual values of the linear regression of the log-transforms of spindle number against muscle mass” was another more suitable way to describe the abundance of MS. We will follow this method in our up-coming research, the primary purpose of this paper was to summarize the current data of MSs in muscles and to analyze whether any correlation exist between number of MSs and muscle mass. The mentioned article (1971) was written in German and the original paper was failed to access (no link). The related data were achieved from newborn human infants (another review article: “Journal of Theoretical Biology 229 (2004) 263–280. DOI: 10.1016/j.jtbi.2004.03.019” had mentioned this issue), so the data were not included in the present study. We revised the text according to your suggestion.
Revision: Line 224-232 of manuscript. “we found another related review article containing information on human muscle MSs [26]: Longissimus capitis, 507.0 (number of MSs); Semispinalis capitis, 619.0; Longissimus dorsi, 1193.0; Obliquus externus abdominis, 569.0; Pectoralis major, 450.0; Iliocostalis, 796.0; Triceps brachii, 520.0; Iliopsoas, 656.0; Adductor magnus, 437.0; Vastus lateralis, 449.0; Gluteus maximus, 629.0. All data presented in this article [26] failed to reference the sources. In addition to references, most human data were quoted from another study (study samples were obtained from a human newborn cadaver), which was written in German. However, we failed to access the full text of the German article. Therefore, the data involved in this article [26] were not presented and analyzed in the present study. We must clarify this issue to ensure this study’s objectivity.”
Specific points:
★L 13 “never” to read “nerve”
Answer: We do apologize. We check the English writing carefully and hand manuscript to MDPI for English editing.
★L 14 abbreviation IMCT not defined
Answer: Sorry for this, it has been revised.
★Lines 44-48 unclear
Answer: Thank you so much, the text has been revised.
Revise: 49-53. “MSs are the principal kinesthetic receptors in mammal skeleton muscles: they inform the central nervous system about muscle fiber length modifications and contribute to motor control and the body’s sense of positioning [9]. MSs are located in almost all muscles and attached to extrafusal muscle fibers in a consistent direction [10]. It has been suggested that there are approximately 50,000 MSs in the entire human musculature [11].”
★L 64 what does “greater biological characteristics” mean?
Answer: Sorry for this error, we revised it.
Revise: Line 71. “γ-motoneurons are the efferents of intrafusal fibers; some intrafusal fibers are also innervated by β-motoneurons, which connect to the extrafusal [13,20].”
★Lines 67-69 no identifiable source is given for the conduction velocities, which are certainly wrong.
Answer: We deleted those contexts.
Revise: Line 72-74. “Axons of efferent and afferent fibers enter the spindle through the outer membrane at the equator region; the efferent motoneurons separate and connect to the polar regions [16,20]. γ-motoneurons can be divided into dynamic and static fibers [20]; dynamic γ fibers are connected to nuclear bag fibers and static fibers are connected to nuclear chain fibers [13,21]. ”
★L 74 what does “the same arrow direction and anatomical characteristics” mean? Intrafusal and extrafusal fibres are certainly not anatomically similar.
Answer: Dear Reviewer, thank you so much for this precious advice, we do make a mistake here, and it has been revised.
Revise: Line 78-79. “The spindles’ intrafusal fibers expand when related muscle fibers are stretched since the spindle intrafusal and muscle fibers run in the same direction [22]. It is well established that action potentials will be generated in afferent sensory neurons according to the degree and rate of stretching”
★Lines 76-79 Primary endings, as noted elsewhere in the paper, usually innervate all three types of intrafusal fibre, and consequently show both dynamic and static sensitivity.
Answer: We are very grateful for the advice. We agree with this, sorry for the inaccuracy here. It has been revised based on your advice.
Revise: Line 81-84.
★L 82 Phospholipase D isn’t an ion channel.
Answer: We are very sorry about this, thank you for pointing out this. It has been deleted.
★L 84 the spinal cord is an integral part of the central nervous system.
Answer: Sorry, the writing here is not rigorous. We deleted it.
★L 98 does PRISMA require explanation and a reference?
Answer: Dear reviewer, we do apologize here, we believe this paper is a narrative review, so we changed the title and followed the guidelines for narrative reviews (Baethge C, Goldbeck-Wood S, Mertens S. SANRA-a scale for the quality assessment of narrative review articles. Res Integr Peer Rev. 2019 Mar 26;4:5.).
Revise: Line 95. “We followed the guidelines for narrative reviews [25].”
★Lines 108 and 136 delete “respectively” to make sense.
Answer: Thank you so much, we are very grateful. We delete it.
Revise: Line 115, 143.
★Lines 151-2 Does “A total of 62 types of 90 muscles” mean “A total of 90 exemplars of 62 different muscles”?
Answer: Dear reviewer, sorry about this, we revised it.
Line 163-164. “Regarding distribution, 56 studies were included. A total of 90 muscles (62 types) were investigated in 20 species,”
★L 170 and elsewhere. Does “pieces of muscle” mean “individual muscles”?
Answer:Thank you so much for pointing this inaccuracy. We revised the sentence to make it clear.
Line 176-177. “The soleus and masseter were the main targets among the total 189 animal muscles (see Supplementary Figure 4E-4F and Supplementary Table S4).”
★In the legend to fig 4 do “results of part 2.1” and “results of part 2.22 refer back to the Methods? If so, there is no 2.2, only 2.1.2. Also F and G are confused.
Answer: This is our fault, please accept our apologies. We have fixed it, and the figure 4 has been uploaded as a supplementary Figure (in order to keep the main text clear and neat).
★Lines 243-245 ‘This proposal seems similar to the original conclusion of “subjective descriptions of fine motor control and brought out the concept that the density of MSs is higher in those muscles which own precious control of movement” [25].’ This appears to be a quotation from ref 25, but the sentence does not occur in that reference.
Answer: Sorry for this, it has been fixed, mistake of reference management.
★References: while this paper was with the reviewer, another relevant paper on the distribution of spindles in human deep cervical muscles has been published by Wang et al. Frontiers in Neuroanatomy https://doi.org/10.3389/fnana.2024.1340468IF: 2.9 Q1
Answer: Dear reviewer, thank you so much. This great article is the most powerful evidence which will support our work. We have quoted it in our manuscript.
★Comments on the Quality of English Language
As indicated above, extensive editing of the English is needed. There are many examples where it is difficult or impossible to even guess at what the authors intend.
Answer: Sorry for the misunderstanding which caused by English writing skills. We check the manuscript carefully and hand it to MDPI for English editing. Certificate has been uploaded to the journal.
Thank you in advance and have a nice day. I am looking forward to hearing from you.
Best regards
All authors
23/6/2024
Reviewer 2 Report
Comments and Suggestions for Authors
The authors have performed a systematic review on distribution muscle spindle (MS) in the muscle. The authors have included 56 papers for distribution of the MS and 44 papers for number of MS. The study is very broad and the authors didnot address any questions related to distribution and number of MS. The figures presented in the study arenot self explanatory. The study is still preliminary in nature. The result and discussion sections arenot written to address the questions of the manuscript. My comments are provided below
1. Please make a table for Figure 4E and 4G.
2. The legend to figure 5 is missing.
3. Please write in details about Figure 6.
4. It will be a good if the authors could reanalyze the whole result and make the figures to address the questions.
5. The manuscript is still missing sufficient evidence for number and distribution of MS in muscle.
6. The authors didnot address the limitation of the study.
7. The animal and human muscle differ a lot, particularly in neuromuscular junction. The authors didnot address this issue.
Comments on the Quality of English LanguageMinor spelling mistakes in the text.
Author Response
Dear Reviewer:
First and foremost, we would like to express our deepest gratitude for your thorough review of our manuscript. Your insightful comments and suggestions are invaluable to us, and we appreciate the time and effort you have dedicated to providing this feedback. We read all carefully and revised our manuscript based on the precious advice. Thank you so much.
The following content in italics (labeled with ★) is your valuable suggestions, text in red is our correction.
The authors have performed a systematic review on distribution muscle spindle (MS) in the muscle. The authors have included 56 papers for distribution of the MS and 44 papers for number of MS. The study is very broad and the authors didnot address any questions related to distribution and number of MS. The figures presented in the study arenot self explanatory. The study is still preliminary in nature. The result and discussion sections arenot written to address the questions of the manuscript. My comments are provided below
★ Please make a table for Figure 4E and 4G.
Answer: Thank you so much, it was our fault not to have thought of it. A table for Figure4e/g is helpful to make reader easier to understand the content. We made a table and up-load to journal as a supplementary table, and the Figure 4 has been re-formed and been up-loaded as a supplementary figure in order to make the main context clear and neat.
★ The legend to figure 5 is missing.
Answer: We do apologize, it has been fixed.
★ Please write in details about Figure 6.
Answer: Dear reviewer, we made another figure in order to make the results and data clearer and easier to understand.
★ It will be a good if the authors could reanalyze the whole result and make the figures to address the questions.
Answer: Thank you, we do miss something, we do believe the picture has great help on readers. We made another figure (Figure 7) also a new figure6.
★ The manuscript is still missing sufficient evidence for number and distribution of MS in muscle.
Answer: We do apology about this. One article (1971) was written in German, and the related data were achieved from newborn human infants (another review article: “Journal of Theoretical Biology 229 (2004) 263–280. DOI: 10.1016/j.jtbi.2004.03.019” had mentioned this issue), so the data were not included in the present study, also the original paper was fail to access (no link). We interpret the reason and showed some details in the main context.
Revision: Line 224-232 of manuscript. “we found another related review article containing information on human muscle MSs [26]: Longissimus capitis, 507.0 (number of MSs); Semispinalis capitis, 619.0; Longissimus dorsi, 1193.0; Obliquus externus abdominis, 569.0; Pectoralis major, 450.0; Iliocostalis, 796.0; Triceps brachii, 520.0; Iliopsoas, 656.0; Adductor magnus, 437.0; Vastus lateralis, 449.0; Gluteus maximus, 629.0. All data presented in this article [26] failed to reference the sources. In addition to references, most human data were quoted from another study (study samples were obtained from a human newborn cadaver), which was written in German. However, we failed to access the full text of the German article. Therefore, the data involved in this article [26] were not presented and analyzed in the present study. We must clarify this issue to ensure this study’s objectivity.”
★ The authors didnot address the limitation of the study.
Answer: We are very sorry about this, our review article has limitations exactly, we did not mention it in the previous draft, and here we have revised it based on the limitations of our work.
Line 407-415. “Limitations remain; even the present study followed the relevant principles in the literature search and data extraction and analysis: the inclusion and exclusion criteria were relatively strict, which might lead to some articles not being included. In particular, the language limitation led to the absence of some non-English articles, which might have an impact on the conclusions of the present study. Second, this study failed to classify and analyze the effects of different MS research methods (staining methods, counting methods, etc.) on the results regarding the number and distribution of MSs. Considering the amount of muscle data available at present, it is not possible to provide a scientific and comprehensive answer to this question.”
★ The animal and human muscle differ a lot, particularly in neuromuscular junction. The authors didnot address this issue.
Answer: We are sorry that we don't realize this problem. We realize that the neuromuscular junction in human and animal is different (the structure and pattern in muscle). We illustrated in the context.
Revised: Line: 300-306. “It is important to note the neuromuscular junction structural differences between animals and humans. A previous study showed that the neuromuscular junction in humans was much smaller and more limited [33]. This difference may suggest that MSs in human muscles are much more concentrated than in animal muscles, but no study has clarified the relationship between the neuromuscular junction and MSs. The present study also failed to clarify this issue.”
★ Comments on the Quality of English Language. Minor spelling mistakes in the text.
Answer: We are sorry about this, we check the manuscript carefully and hand it to MDPI for English editing. Certificate has been uploaded to the journal.
Thank you in advance and have a nice day. I am looking forward to hearing from you.
Best regards
All authors
23/6/2024
Round 2
Reviewer 1 Report
Comments and Suggestions for Authors
The authors have satisfactorily addressed most of my comments on the original version, though I don't think it is helpful to continue to quote spindle densities. The original data would, of course, have been the total numbers of spindles in each muscle and the weight (or mass) of the adult muscle. These data are given in figure 6 and in my opinion have primacy and are sufficient. I will not press the point further, however, as it is certainly useful to have so much data collected together.
I still have some specific points, now that the English has been much improved:
L 47 “mammal skeleton muscles” should read “mammalian skeletal muscles”
L 52 “MS are spindle-shaped since intrafusal fibers expand in diameter..” No – muscle spindles are spindle shaped because the capsule increases in diameter where it encloses the sensory ending(s); the intrafusal fibres usually attain their greatest diameters in the polar regions.
L 57-58 “There are three types of intrafusal fibers: large nuclear bag1 fibers, larger nuclear bag2 fibers, and smaller nuclear chain fibers.” As the review also includes avian spindles, it should be pointed out that the three types of intrafusal fibre listed here are only found in mammalian spindles.
L 62-63 “the type Ia and type II nerve fibers are the dominant sensory innervation”. Once again, this only applies to mammalian spindles.
L 66-67 “Type II nerve terminals are embedded in the side of type Ia fibers’ annulospiral ending”. I assume this means that the secondary endings are located on either side of the primary ending.
L 72-73 “dynamic γ fibers are connected to nuclear bag fibers and static fibers are connected to nuclear chain fibers”. Actually dynamic γ axons innervate bag1 fibres, static γ axons innervate bag2 and chain fibres.” But only in mammals.
L 75 “The spindles’ intrafusal fibers expand when…” should probably read “The spindles’ intrafusal fibers extend when…”
L 218 “this study” Does this refer to the present review, or to reference 26?
L 230-232 “This proposal seems similar to the original hypothesis that ‘muscles with high densities are primarily involved in fine motor control’ [27]” The full context of the quotation is, however: “It is a commonly held belief that the provision of muscle spindles reflects the functional demands of a given muscle, with some hypothesising that muscles with high spindle densities (number of muscle spindles per gram) are primarily involved in fine motor control or function as kinesiological sensors. There are, however, several fundamental issues with this hypothesis."
Comments on the Quality of English LanguageThe English is greatly improved from the original version, and the paper is now clear more or less throughout.
Author Response
Dear Reviewer:
We sincerely appreciate the detailed and thoughtful feedback you provided on our revised manuscript. Without you invaluable assistance, the article would not exist in its present form. Your constructive comments and suggestions have prompted us to make significant improvements. We have addressed all the points raised and believe that the revisions have substantially strengthened the manuscript. Thank you once again for your time and valuable contributions to our research.
The following content in italics (labeled with ★) is your valuable suggestions, text in red is our correction. We also highlight the revision in the manuscript whit green background.
★The authors have satisfactorily addressed most of my comments on the original version, though I don't think it is helpful to continue to quote spindle densities. The original data would, of course, have been the total numbers of spindles in each muscle and the weight (or mass) of the adult muscle. These data are given in figure 6 and in my opinion have primacy and are sufficient. I will not press the point further, however, as it is certainly useful to have so much data collected together.
Answer: Dear Reviewer. We express our appreciation. This paper would not have reached its current form without your insightful advice. We apologize for not conducting additional analysis to clarify the abundance according to the method of "residual values of the linear regression of the log-transforms of spindle number against muscle mass." The current content represents the initial phase of our research, and we plan to conduct this analysis in our subsequent study, as per your recommendation. Thank you once again for your guidance and understanding.
★I still have some specific points, now that the English has been much improved:
Answer: Your encourage means a lot to us, thank you.
★ L 47 “mammal skeleton muscles” should read “mammalian skeletal muscles”
Answer: Sorry about this, we revised it to right form of “mammalian skeletal muscles”.
★L 52 “MS are spindle-shaped since intrafusal fibers expand in diameter..” No – muscle spindles are spindle shaped because the capsule increases in diameter where it encloses the sensory ending(s); the intrafusal fibres usually attain their greatest diameters in the polar regions.
Answer: Thank you so much, it has been revised.
Line 55-56: “MS are spindle-shaped. In the central region of the muscle spindle, sensory nerve terminals are observed to connect with intrafusal fibers, and the diameter of spindle capsule increased in this area”
★L 57-58 “There are three types of intrafusal fibers: large nuclear bag1 fibers, larger nuclear bag2 fibers, and smaller nuclear chain fibers.” As the review also includes avian spindles, it should be pointed out that the three types of intrafusal fibre listed here are only found in mammalian spindles.
Answer: Dear Reviewer. We apologize for the inaccuracy, it has been revised. We limited it in mammalian spindles.
Line: 62: “There are three types of intrafusal fibers in mammalian spindles: large nuclear bag1 fibers, larger nuclear bag2 fibers, and smaller nuclear chain fibers.”
★L 62-63 “the type Ia and type II nerve fibers are the dominant sensory innervation”. Once again, this only applies to mammalian spindles.
Answer:It has been revised. We limited it in mammalian spindles.
Line: 66-67: “Mammalian MSs have afferent and efferent nerve integrated innervation; the type Ia and type II nerve fibers are the dominant sensory innervation”
★L 66-67 “Type II nerve terminals are embedded in the side of type Ia fibers’ annulospiral ending”. I assume this means that the secondary endings are located on either side of the primary ending.
Answer: Sorry for the mistake, we do apologies and revised it: Line 70.
★L 72-73 “dynamic γ fibers are connected to nuclear bag fibers and static fibers are connected to nuclear chain fibers”. Actually dynamic γ axons innervate bag1 fibres, static γ axons innervate bag2 and chain fibres.” But only in mammals.
Answer: Thank you so much, the advice will make the paper much more accurate and valuable. It has been revised; we indicated at the beginning of the paragraph that it was mammalian spindles.
Line 76-77: “dynamic γ fibers are connected to nuclear bag 1 fibers and static fibers are connected to bag 2 and chain fibers”
★L 75 “The spindles’ intrafusal fibers expand when…” should probably read “The spindles’ intrafusal fibers extend when…”
Answer: Dear Reviewer. Sorry for the inaccuracy, it has been revised. Line 79.
★L 218 “this study” Does this refer to the present review, or to reference 26?
Answer: Sorry for the confusion, it is the reference 26. We revised it to make it clear.
Line 218-219: “Data presented in the article (the related review article, reference No.26) failed to reference the sources.”
★L 230-232 “This proposal seems similar to the original hypothesis that ‘muscles with high densities are primarily involved in fine motor control’ [27]” The full context of the quotation is, however: “It is a commonly held belief that the provision of muscle spindles reflects the functional demands of a given muscle, with some hypothesising that muscles with high spindle densities (number of muscle spindles per gram) are primarily involved in fine motor control or function as kinesiological sensors. There are, however, several fundamental issues with this hypothesis."
Answer: Dear Reviewer. Sorry for the inaccuracy, we misunderstand the point of the authors, it has been revised: Line 241: “Another study [27] about muscle architecture and MS abundance also expressed reservations regarding above hypothesis.”
Thank you so much for reviewing our manuscript. Have a nice day!
Best regard
All authors
27/6/2024
Reviewer 2 Report
Comments and Suggestions for Authors
The authors have addressed all my concerns in the revised version. I support the publication of the manuscript.
Comments on the Quality of English LanguageMinor spelling mistakes are detected in the text.
Author Response
Dear Reviewer:
We sincerely appreciate the detailed and thoughtful feedback you provided on our revised manuscript. Without you invaluable assistance, the article would not exist in its present form. Your constructive comments and suggestions have prompted us to make significant improvements. Thank you once again for your time and valuable contributions to our research.
We have reviewed the entire text and found some spelling errors, which we have now corrected, and we contact the author service center of MDPI because present manuscript is revised by the English editing service from MDPI. Thank you so much for your help and have a nice day.
Best regards
All authors
27/6/2024